# Contribution of Intrinsic Fluorescence to the Design of a New 3D-Printed Implant for Releasing SDABS

**DOI:** 10.3390/pharmaceutics12100921

**Published:** 2020-09-26

**Authors:** Alexandre Nicolas, Alice Dejoux, Cécile Poirier, Nicolas Aubrey, Jean-Manuel Péan, Florence Velge-Roussel

**Affiliations:** 1GICC EA 7501, Faculty of Medicine, University of Tours, 37032 Tours, France; alexandre.nicolas@univ-tours.fr; 2PEX DPH, Technologie Servier, 45000 Orleans, France; a.dejoux19@imperial.ac.uk (A.D.); cecile.poirier@servier.com (C.P.); jean-manuel.pean@servier.com (J.-M.P.); 3ISP UMR 1282, INRA, Team BioMAP, University of Tours, 37200 Tours, France; nicolas.aubrey@univ-tours.fr

**Keywords:** protein formulation(s), nanobody/V_H_H, scFv, bispecific antibody, controlled release, intrinsic fluorescence, thermo-responsive polymers, implant, 3D printing

## Abstract

Single-domain antibodies (sdAbs) offer great features such as increased stability but are hampered by a limited serum half-life. Many strategies have been developed to improve the sdAb half-life, such as protein engineering and controlled release systems (CRS). In our study, we designed a new product that combined a hydrogel with a 3D-printed implant. The results demonstrate the implant’s ability to sustain sdAb release up to 13 days through a reduced initial burst release followed by a continuous release. Furthermore, formulation screening helped to identify the best sdAb formulation conditions and improved our understanding of our CRS. Through the screening step, we gained knowledge about the influence of the choice of polymer and about potential interactions between the sdAb and the polymer. To conclude, this feasibility study confirmed the ability of our CRS to extend sdAb release and established the fundamental role of formulation screening for maximizing knowledge about our CRS.

## 1. Introduction

The advent of therapeutic proteins lies in their high specificity and complex set of functions. The evolution of molecular engineering and phage display technology has made treatment with therapeutic proteins possible. From January 2014 to July 2018, almost 150 new products were approved by the Federal Drug Administration (FDA) and/or the European Medicines Agency (EMA) [1]. The predominant therapeutic proteins on the market are immunoglobulins (Ig) and more specifically IgG1 [2]. They are principally used for treating autoimmune, cardiovascular, and infectious diseases as well as cancer and inflammation [1]. Antibodies (Abs) can play a variety of roles, such as blocking cell receptors, inducing target cell cytotoxicity, and acting as a carrier for potent small molecules [3]. Abs display a high specificity, an extended serum half-life, and immune effector functions [4]. This extended circulation time of IgG is a great advantage. It is mainly mediated by molecular weight (~150 kDa) of IgG and their ability to interact with the neonatal Fc receptor (FcRn). Indeed, the molecular weight of IgG prevents the kidneys from eliminating it by excretion through the glomerulus [5,6]. In contrast to others therapeutic proteins, Ab treatments only require injections once every 2–3 weeks. Furthermore, Abs are characterized by limited penetration into solid tumors [7,8]. Ab extravasation from systemic blood circulation to target tissues is principally mediated by convective flow and, to a lesser degree, by diffusion [9]. Ab penetration is notably limited by their molecular weight, which led to the advent of single domain antibodies (sdAbs) [10]. Therefore, Ab development is not straightforward, with technological challenges such as protein solubility, stability, distribution, and route of administration [11,12]. However, more than these technical challenges, the evolution of protein therapeutics is driven by innovations to meet competition from other biopharmaceutical products (gene therapy and CAR-T cells) and deal with a crowded market. These innovations focus on the development of enhanced binding domains, multi-specificity, Fc engineering, and chemical engineering to link proteins to highly potent active pharmaceutical ingredients (APIs) [1]. An answer to the limited penetration of IgG was the use of sdAbs. These still have high specificity since the variable domains of the whole antibody are conserved, but they are characterized by a drastic reduction in molecular weight. The three main types of sdAbs are antigen-binding fragments (Fab), single-chain variable fragments (scFv), and nanobodies [13]. The small molecular weight of sdAbs favors their penetration into inaccessible tissues, with Fab and scFv penetration having been shown to be largely superior to that of intact IgG [14,15]. Furthermore, the absence of the Fc domain allows sdAbs to be produced using more cost-effective methods, such as prokaryotic systems. sdAbs have demonstrated almost identical specificity and affinity but greater stability and solubility. sdAbs could theoretically be administrated at one-tenth of the dose of an equivalent IgG therapeutic, thus reducing solution viscosity [16]. Some sdAbs, such as nanobody also named V_H_H, are able to reversibly unfold after thermal denaturation [17]. Nonetheless, all these benefits should be weighed against one of the main drawbacks of sdAbs, namely that they have a very short serum half-life due to this absence of the Fc domain. For example, studies have shown sdAbs to be rapidly excreted by the kidneys and metabolized or reabsorbed by the proximal tubes [18].

As stated above, the evolution of biopharmaceuticals is driven by innovations. This being so, different strategies were studied to overcome the limited half-life of sdAbs. These strategies mainly focused on protein engineering and drug delivery systems (DDS). DDS, and more particularly controlled release systems (CRS), are particularly attractive because they enable long-lasting local drug concentrations, improved efficacy-dosing ratios, and reduced treatment-associated side effects. Numerous CRS have been advanced to extend protein release, including microparticles, nanoparticles, implants, depot systems, and hydrogels. Hydrogel is a tridimensional matrix created by physical or chemical crosslinking, and it is regularly used to sustain protein delivery because of their biocompatibility and their ability to control protein release [19]. Thermo-responsive hydrogels are frequently made of poloxomer 407 (P407), also called Pluronic F-127, and poly(N-isopropylacrylamide), also called pNIPAAM. P407 is a triblock copolymer comprising blocks of ethylene oxide and propylene oxide, while pNIPAAM is a homopolymer comprising repeating blocks of N-isopropylacrylamide. The main advantage of thermo-responsive polymers is their ability to remain liquid at room temperature, thus ensuring injectability, and to form a reversible tridimensional matrix at body temperature [20,21]. One other well-known biomaterial is alginate, a natural anionic polymer extracted from brown seaweed that forms a tridimensional matrix in the presence of divalent cations. It is a copolymer that consists of linear blocks of guluronate and manuronate, with only the guluronate blocks interacting with the divalent cations [22]. Another common way of sustaining protein release is entrapping the protein inside an implant. Such implants were initially developed to overcome problems with oral therapy and are now used to improve safety, efficacy, and patient compliance [23]. They can now be produced by 3D printing due to the adaptability of this technology, which also reduces costs [24].

In this study, we explored a new CRS that combines the novelty of a 3D-printed implant with the robustness of well-established hydrogel formulations. The hydrogel was used to control sdAb release and was combined with the implant to control surface exchange between the hydrogel and the release medium. This combination with the implant also permitted an easy removal of the device if adverse effects occurred. Through this feasibility study, we explored the delivery of different sdAbs such as V_H_H and bispecific tandem scFv using our new 3D-printed implant. This work aimed to demonstrate the ability of our concept to extend Ab fragment release up to nearly two weeks through an initial burst release followed by a continuous release. Our results point out the fundamental role of preliminary screening, including intrinsic fluorescence experiments, to establish the best formulation conditions and the most appropriate polymer effect on protein tertiary structure.

## 2. Materials and Methods

### 2.1. Materials

Citric acid, sodium phosphate monobasic dihydrate, sodium phosphate dibasic dihydrate, calcium chloride, sodium chloride, P407, and pNIPAAM were purchased from Sigma-Aldrich (Saint-Quentin-Fallavier, France). High-viscosity alginate sodium salt derived from brown algae (viscosity: 1320 mPa.s 1% *w/v*) and a Pierce™ Micro BCA™ kit were obtained from Thermo Fisher Scientific (Illkirch-Graffenstaden, France). Transparent polylactic acid (PLA) was obtained from Makershop (Le Mans, France). Milli-Q grade water (18.2 MΩ.cm) was used to prepare all solutions.

### 2.2. Protein Solutions

A bispecific tandem scFv against CLEC4A and TLR-2 (BIC25) was produced in GibcoTM ExpiCHO-STM cells from Thermo Fisher Scientific (Illkirch-Graffenstaden, France). The vector was a pCDNA3.4 with a promotor CMV. BIC25 has a theoretical molecular weight of 53.2 kDa and an isoelectric point of 6.23. BIC25 was diluted to 0.5 mg/mL in 50 mM borate buffered saline and 150 mM sodium chloride, and stored at −80 °C.

A nanobody against EGFR1 known as anti-EGFR1 V_H_H (V_H_H_EGFR1_) was produced in a soluble form in a BL21 strain of *Escherichia coli* from New England Biolabs (Evry, France) and supplied by Selvita (Kraków, Poland). The vector was a pET-15b with a promotor T7. V_H_H_EGFR1_ has a theoretical molecular weight of 15.3 kDa and an isoelectric point of 6.57. V_H_H_EGFR1_ was diluted to 2 mg/mL in 50 mM Tris-HCl, 300 mM sodium chloride, and 5% glycerol (*v*/*v*) and stored at −80 °C.

A nanobody against HER2 known as anti-HER2 V_H_H (V_H_H_HER2_) was produced as inclusion bodies in a BL21 strain of *E.coli* from New England Biolabs (Evry, France) and refolded through buffer exchange, as described in a previous report [25]. The vector was a pET-15b with a promotor T7. V_H_H_HER2_ has a theoretical molecular weight of 14.2 kDa and an isoelectric point of 9.23. V_H_H_HER2_ was diluted to 2 mg/mL in 25 mM Tris-HCl and 150 mM sodium chloride and stored at −80 °C.

### 2.3. Sample Preparation

#### 2.3.1. Preparation of Citrate–Phosphate Solutions

Buffered solutions were prepared to obtain pH values of 6.0, 7.0, and 8.0, and adjusted ionic strengths of 50, 150, and 300 mM. The solution’s pH was adjusted using 0.1 M citric acid and 0.2 M sodium phosphate dibasic dihydrate, while its total ionic strength was adjusted using sodium chloride. The contribution of the citrate–phosphate to the solution’s total ionic strength was fixed at 25 mM according to the Henderson–Hasselbach equation.

#### 2.3.2. Preparation of sdAb Solutions

The sdAbs were concentrated to 3 mg/mL using the Amicon^®^ Ultra-0.5 centrifugal filter from Sigma-Aldrich (Saint-Quentin-Fallavier, France) and then dialyzed at 4 °C against the citrate–phosphate solutions using the Pierce™ 96-well Microdialysis Plate (3.5K MWCO) from Thermo Fisher Scientific (Illkirch-Graffenstaden, France). The ratio of protein to buffer was 1:16 (*v*/*v*), at 100 µL of protein solution to 1.6 mL of dialysis buffer. The dialysis buffer was exchanged 3, 6, 16, 19, 22, and 32 h after dialysis began. The protein solutions and dialysis buffers were recovered and sterile-filtered under a laminar flow cabinet using 0.22 µm polyethersulfone filters from Sigma-Aldrich (Saint-Quentin-Fallavier, France). The sdAb solutions were centrifuged for 20 min at 20,000 *g* and 4 °C and then sterile-filtered as previously described. Protein concentration was determined by measuring absorption at 280 nm using a NanoDrop^TM^ 100 from Thermo Fisher Scientific (Illkirch-Graffenstaden, France). The protein solutions were adjusted to 0.5 and 0.2 mg/mL for subsequent biophysical studies.

#### 2.3.3. Preparation of Hydrogels

The alginate, P407, and pNIPAAM solutions were prepared by dissolving each polymer in a citrate–phosphate solution with the last two in an ice bath. Polymer was gradually added under magnetic stirring until apparent solution homogeneity was achieved. The final solutions were kept at 4 °C for 48 h under slight magnetic stirring until the polymer fully dissolved. The concentrations of the pNIPAAM and P407 solutions were 10 and 250 mg/mL, respectively. The sodium alginate was dissolved in a citrate–phosphate solution at 40 °C. Alginate was added to the citrate–phosphate solution under gentle magnetic stirring. The resulting 10 mg/mL solutions were then magnetically stirred at room temperature overnight. Finally, the pNIPAAM, alginate and P407 solutions were kept at 4 °C for 24 h. The sdAb solutions were concentrated up to 6.5 mg/mL using the Amicon^®^ Ultra-0.5 centrifugal filter from Sigma-Aldrich (Saint-Quentin-Fallavier, France). The sdAb and polymer solutions were mixed at a ratio of 1:4 (*v*/*v*). The resulting concentrations of sdAb, pNIPAAM, alginate, and P407 were 1.5, 8, 8, and 200 mg/mL, respectively. To improve the solutions’ homogeneity, they were mixed by means of multiple pipetting, left at 4 °C for 24 h, pipetted a second time, and then stored at 4 °C.

### 2.4. Thermal Stability of sdAbs

#### 2.4.1. Intrinsic Fluorescence Measurements

Intrinsic fluorescence assays were performed with the Prometheus NT.48 from NanoTemper Technologies GmbH (Munich, Germany). Analysis was performed on 10 µL of each sdAb formulation, which was introduced into standard capillaries from NanoTemper Technologies GmbH (Munich, Germany). Thermal denaturation was obtained by scanning from 20 to 95 °C and from 95 to 20 °C at a heating rate of 1 °C/min. The excitation wavelength was set at 280 nm, and fluorescence intensity was recorded at wavelengths of 330 and 350 nm. Our results were plotted as a ratio of fluorescence intensity at 350 or 330 nm to temperature. All fitted values were visually inspected to eliminate possible artifacts. All curves were analyzed with GraphPad Prism software and the mean spectrum was reported. The parameters we analyzed were initial fluorescence intensity ratio (350/330i), apparent melting temperature (Tmapp), variation in apparent melting temperature according to concentration (ΔTmapp), and variation in mean ratio from 20 to 22 °C between heating and cooling (Δratio). The 350/330i was calculated as the mean fluorescence ratio between 20 and 22 °C during the heating phase. Tmapp was determined by Boltzmann fitting of the fluorescence ratio curve using GraphPad Prism. ΔTmapp was calculated as the difference in Tmapp values at 0.5 and 0.2 mg/mL, as previously described [26]. Finally, Δratio was the absolute difference in mean fluorescence ratio from 20 to 22 °C between heating and cooling ramp [26]. The influence of pH and ionic strength was analyzed in the sdAbs at 0.5 mg/mL, and in citrate–phosphate solution at pH 6.0, 7.0, and 8.0 and at ionic strengths of 50, 150, and 300 mM. This analysis was also performed on V_H_H_EGFR1_ at 0.2 mg/mL. The influence of the polymers was analyzed in the sdAbs at 0.5 mg/mL in the presence of pNIPAAM, alginate, and P407 and in citrate–phosphate solution at pH 6.0, 7.0, and 8.0 and at ionic strengths of 50, 150, and 300 mM. This analysis was also performed in V_H_H_EGFR1_ at 0.2 mg/mL in the presence of pNIPAAM.

#### 2.4.2. Backscattering Measurements

In addition to the intrinsic fluorescence assays discussed above, backscatter analyses were performed with a Prometheus NT.48 from NanoTemper Technologies GmbH (Munich, Germany). Scatter signal during heating was recorded using back-reflection optics. Our results were expressed as scatter signal according to temperature. All curves were analyzed with GraphPad Prism and the mean spectrum was reported. Additionally, all fitted values were visually inspected to eliminate possible artifacts.

### 2.5. TRelease Study of V_H_H_HER2_

#### 2.5.1. 3D Printing of the PLA Implant

The implants were fabricated using commercially available transparent Ultimaker PLA and the Ultimaker 3 3D printer from Makershop (Le Mans, France). Print runs were performed with an Ultimaker 0.25 mm AA print core from Makershop (Le Mans, France). The geometry of the implant was designed using Fusion 360 software from Autodesk (San Rafael, CA, USA). The file generated was exported as a stereolithography file into the CURA 3.5 software from Ultimaker (Le Mans, France).

The implant was a hollow cylinder with an external diameter of 3.3 mm, an internal diameter of 2.3 mm, and a length of 32 mm. It was designed to have only one end open to the external environment. The printing parameters were layer height of 0.1 mm, printing speed of 20 mm/s, movement speed of 250 mm/s, 100% infill, 185 °C nozzle temperature, and 60 °C build temperature. The implants were stored at room temperature in a closed box which contained desiccant sachets.

#### 2.5.2. Scanning Electron Microscopy (SEM)

The surface morphology of the PLA implants was studied using the MIRA-3 FEG scanning electron microscope from TESCAN (Fuveau, France). The analysis was carried out at a pressure of 15 Pa and a voltage of 25 kV. The implant was cut to observe its interior, exterior, and open end.

#### 2.5.3. Differential Scanning Calorimetry

Differential scanning calorimetry (DSC) was performed to assess the thermal behavior of the PLA implants. The assays were performed with a conventional DSC 822e from METTLER-TOLEDO (Viroflay, France). The PLA implants (4 mg) were sealed in a 40-µL aluminum pan and heated at a scan rate of 10 °C/min from 0 to 220 °C, with nitrogen being used as a purging gas (50 mL/min). Thermograms were analyzed with the STARe software V9.10 from METTLER-TOLEDO (Viroflay, France). The parameters we analyzed were the glass temperature transition (Tg) and melting temperature (Tm).

#### 2.5.4. Loading of Implants with Hydrogel

The ability of the hydrogel-loaded implants to sustain protein release was assessed by measuring V_H_H_HER2_ release from alginate and P407 hydrogel. Implant filling and all subsequent operations were performed under a laminar flow cabinet. Furthermore, the implants were handled with 70° tweezers which had been disinfected with alcohol. After the polymer or sdAb solution had been removed from 4 °C storage, the implant was quickly filled using a 1 mL syringe and a 0.8 mm × 50 mm needle from TERUMO (Leuven, Belgium). The volume of alginate and P407 solution loaded into the implants was 135 µL. A particularity of the alginate-loaded implants was that 25 µL of the alginate solution was removed and replaced by 25 µL of 0.1 M calcium chloride to trigger hydrogel crosslinking. The preparation of the V_H_H_HER2_ and polymer solutions is described above under “Preparation of hydrogels”. The initial concentrations of V_H_H_HER2_, alginate, and P407 were 1.5, 8, and 200 mg/mL, respectively. The pH of the hydrogels was 6.0, 7.0, and 8.0 and their ionic strength was set at 50 mM. Controls were also prepared which consisted of implants loaded with hydrogel that did not contain V_H_H_HER2_.

#### 2.5.5. Hydrogel Homogeneity (Intrinsic Fluorescence)

The hydrogel homogeneity was analyzed with the help of antibody fragment’s fluorescence. Intrinsic fluorescence assays were performed with standard capillaries (10 µL) and the Prometheus NT.48 from NanoTemper Technologies GmbH (Munich, Germany). The mean fluorescence intensity at 330 nm (F330i) of each antibody fragment was measured between 20 and 22 °C in solution, which contained pNIPAAM, alginate, and P407. The homogeneity of antibody fragment within each polymer solution was assessed through the variation of the mean fluorescence intensity between each sampling.

#### 2.5.6. Release Study (BCA Assay)

The release study was performed using a Wheaton vial from VWR (Fontenay-sous-Bois, France), which was filled with 4.3 mL of 50 mM sodium phosphate at pH 7.5 and 37 °C and 150 mM sodium chloride. The vial containing the release medium was heated to 37 °C for 30 min before the implant was introduced. The hydrogel-loaded implant was placed in the preheated vial, which was in turn placed in an Innova 4000 incubator shaker from New Brunswick Scientific (Enfield, CT, USA) set at 37 °C and an agitation speed of 25 rpm. At predetermined time intervals of 24, 144, and 312 h, the implants were transferred into new vials containing fresh release medium. V_H_H_HER2_ concentration in the release medium was measured using a bicinchoninic acid (BCA) colorimetric assay from Thermo Fisher Scientific. The plate was incubated at 37 °C for 2 h before the color change mediated by the BCA due to copper reduction was measured at 562 nm. Absorbance was read using an Epoch™ 2 Microplate spectrophotometer from BioTek Instruments (Winooski, VT, USA). Unknown V_H_H_HER2_ concentrations in the release medium were interpolated from the calibration curve. Blank mean absorbance was subtracted from the reading of each corresponding condition. The calibration curve and samples were prepared and measured in triplicate and duplicate, respectively. Our results were expressed as the mean and standard deviation of each release condition.

#### 2.5.7. Thermal Stability of Released V_H_H_HER2_

Fluorescence spectroscopy was used to assess the tertiary structure of the V_H_H_HER2_ released. Intrinsic fluorescence assays were performed with the Prometheus NT.48 from NanoTemper Technologies GmbH (Munich, Germany). At predetermined time intervals, 10 µL of the release medium of each duplicate were introduced into standard capillaries from NanoTemper Technologies GmbH (Munich, Germany).

The excitation wavelength was set at 280 nm for tryptophan and tyrosine excitation. Fluorescence intensity was measured at wavelengths of 330 and 350 nm. Thermal denaturation was obtained by scanning from 20 to 95 °C and from 95 to 20 °C at a heating rate of 1 °C/min. Fluorescence intensity was recorded at 330 and 350 nm. All fitted values were visually inspected to eliminate possible artifacts. To assess changes in the structure of V_H_H_HER2_, Δratio was used to quantify unfolding reversibility. This parameter was the absolute difference between the mean fluorescence ratios for the initial and final 2 °C of the heating and cooling ramps, respectively.

## 3. Results

### 3.1. Analysis of Polymer Effect on sdAb Thermal Stability

#### 3.1.1. Polymer Effect on BIC25 Thermal Stability

The objective of this thermal stability study was to determine polymer effect on BIC25 tertiary structure. Control experiments were performed in the absence of polymer to distinguish and understand the influence of polymers, solution pH and ionic strength.

In the absence of polymer, the F350/F330 thermograms for BIC25 showed clear fluorescence transitions in the 50–65 °C range (Figure 1A, left). These fluorescence thermograms highlighted the strong influence of the solution’s pH, and to a lesser extent of its ionic strength, on BIC25. Moreover, the light scattering thermogram for BIC25 revealed only one steep rise between 50 and 70 °C (Figure 1A, middle).

This suggests that BIC25 strongly aggregated upon heating between 55 and 70 °C. Additionally, the cooling thermogram was relatively flat in the 20–95 °C range, suggesting no recovery of BIC25 tertiary structure upon cooling (Figure 1A, right). In the presence of polymer, the F350/330 thermograms for BIC25 showed clearly defined transitions (Figure 1B). The F350/330 thermogram of BIC25 seemed to flatten out slightly when pNIPAAM was added (Figure 1B, left). The shape of the BIC25 curve in the presence of alginate was similar to that of the BIC25 solutions without polymer (Figure 1B, middle). In contrast, a noticeable change in BIC25 F350/330 was observed in the presence of P407, namely a considerable flattening of the signals.

Tmapp was used to further investigate the influence of the solution’s environment on BIC25 thermal stability. In the absence or presence of polymer, altering the pH had a strong influence on Tmapp, whereas ionic strength did not exhibit any significant effect. Indeed, a rise in pH from 6 to 8 caused a sharp decrease in Tmapp (Figure 1C, left and right). Of the solutions that contained a polymer, the highest Tmapp value was 61.1 ± 0.1 °C, and it was obtained in pNIPAAM and alginate solutions at pH 6 and 50 mM.

#### 3.1.2. Polymer Effect on V_H_H_EGFR1_ Thermal Stability

As with BIC25 screening, control experiments were performed in the absence of polymer. The F350/330 thermogram for V_H_H_EGFR1_ demonstrated a sharp drop of the fluorescence values between 65 and 75 °C upon heating (Figure 2A, left).

Moreover, the light scattering thermogram displayed two main transitions at 40 and 70 °C during the course of the ramp (Figure 2A, middle). The first fluorescence transition was practically unaffected by pH and ionic strength, whereas the second was. Finally, the F350/330 thermogram for V_H_H_EGFR1_ was relatively flat upon cooling (Figure 2A, right).

The F350/330 thermograms for V_H_H_EGFR1_ in the presence of the polymers are presented in Figure 2B. Firstly, the pNIPAAM evidently altered the F350/330 thermogram of V_H_H_EGFR1_ with the appearance of an additional transition near 40 °C (Figure 2B, left). Secondly, the F350/330 thermograms for V_H_H_EGFR1_ in the presence of the alginate seemed to be slightly changed (Figure 2B, middle). Indeed, the solution pH and ionic strength appeared to strongly influence the alginate effect on V_H_H_EGFR1_ thermal stability. Thirdly, the effect of P407 on the F350/330 thermogram of V_H_H_EGFR1_ was evidently marked as the main fluorescence transition occurred at lower temperature and an additional increase in F350/330 was observed at 80 °C that was not seen in the other solutions (Figure 2B, right).

In the absence of polymer, a rise in pH from 6 to 8 led to a slight increase in Tmapp (Figure 2C, left). In the presence of the polymers, V_H_H_EGFR1_ Tmapp was relatively altered (Figure 2C, middle), a sharp decrease in Tmapp being observed in this setting. The pNIPAAM and alginate solutions were associated with higher Tmapp values. Tmapp increased when the pH rose from 6 to 8. We noted that ionic strength had a slight influence on Tmapp in alginate solutions at pH 6 and 7. Increasing ionic strength from 50 to 300 mM resulted in a substantial increase in V_H_H_EGFR1_Tmapp. As with alginate, the intensity of the P407 effect appeared to correlate with ionic strength, since a rise in ionic strength from 50 to 300 mM resulted in a drop in Tmapp except for the condition P407 at pH 7 and 300 mM. Finally, the highest Tmapp value (70.9 ± 0.1 °C) was observed in alginate solution at pH 8 and 50 mM.

To refine our assessment, ΔTmapp was calculated to gain knowledge about the polymer effect, pH, and ionic strength on V_H_H_EGFR1_ thermal stability. In the absence of pNIPAAM, a pH increase from 6 to 8 led to a slight drop in ΔTmapp value, as shown in Figure 2C (right). Moreover, increasing ionic strength at pH 6 and 7 from 50 to 150 and 300 mM correlated with an apparent increase in ΔTmapp. In the presence of pNIPAAM, increasing the pH from 6 to 8 led to a drop in ΔTmapp. This increase caused ΔTmapp to shift from positive to negative values in solutions at pH 7 and 8. Consequently, the pNIPAAM solutions at pH 6 were associated with higher V_H_H_EGFR1_ Tmapp values at 0.5 mg/mL than at 0.2 mg/mL. When pH decreased from 8 to 6, the influence of ionic strength drastically increased.

#### 3.1.3. Polymer Effect on V_H_H_HER2_ Thermal Stability

As with previous sdAbs, control experiments were performed in the absence of polymer. The F350/F330 thermogram for V_H_H_HER2_ was almost not modified by the solution pH and ionic strength upon heating (Figure 3A, left). Additionally, the light scattering thermogram is relatively flat in the 20–95 °C range upon heating except at pH 8 (Figure 3A, middle). Moreover, the cooling F350/330 thermogram of V_H_H_HER2_ displayed a clear transition during the cooling ramp (Figure 3A, right). The magnitude of this transition mainly related to the pH of the V_H_H_HER2_ solutions, with increased pH resulting in a smaller transition. In the presence of polymer, the F350/330 thermogram for V_H_H_HER2_ was practically unaltered (Figure 3B).

The F350/F330 thermogram in the presence of pNIPAAM was relatively similar to V_H_H_HER2_ solutions without polymer but a potential second transition was observed near 55 °C (Figure 3B, left). The effect of alginate on the V_H_H_HER2_ F350/F330 thermogram clearly showed that alginate heightened the influence of the solution’s ionic strength, particularly at low strengths (Figure 3B, middle). Finally, in the presence of P407, the F350/330 thermogram was strongly reshaped, becoming noticeably more linear and comprising two transitions (Figure 3B, right).

In the absence of polymer, pH and ionic strength had little effect on the Tmapp values of V_H_H_HER2_ (Figure 3C, left), whereas the presence of certain polymers did significantly affect them (Figure 3C, middle). Hence, pNIPAAM and alginate solutions exhibited the highest Tmapp values for V_H_H_HER2_. The P407 solutions, meanwhile, exhibited the lowest. They were more greatly influenced by the solution’s ionic strength, with an increase in this being associated with a sharp decrease in Tmapp. On the contrary, the ionic strength of the pNIPAAM solutions had little impact on Tmapp except at pH 7 and 8, where increasing the ionic strength caused a minor drop in Tmapp. The ionic strength of the alginate solutions only affected Tmapp at 50 mM, at which level it substantially decreased it. In addition, the effect of pH on Tmapp was predominant in the alginate and pNIPAAM solutions: increasing the pH from 6 to 8 led to a significant increase in Tmapp. On the contrary, P407 solutions were practically unaffected by the pH of the buffer. Finally, the highest Tmapp was 69.3 ± 0.2 °C, and it was observed in the pNIPAAM solution at pH 7 and 50 mM. In the alginate solutions, the highest Tmapp value was 69.2 ± 0.1 °C, and it was observed at pH 7 and 300 mM.

Due to the presence of a clear inflection point upon cooling, the reversibility of V_H_H_HER2_ unfolding was assessed using the variation in the ratio between the unfolding and refolding steps (Δratio). In the absence of polymer, the Δratio of V_H_H_HER2_ largely rose as pH increased from 6 to 8 (Figure 3C, right). The lowest value of Δratio for V_H_H_HER2_ was obtained at pH 6. Additionally, an increase in ionic strength at pH 7 and 8 correlated with a drop in Δratio. In the presence of pNIPAAM, the Δratio for V_H_H_HER2_ was strongly impacted by the solution pH. Furthermore, a strong effect of the ionic strength of the pNIPAAM solutions at pH 7 and 8 was noticed as increasing the ionic strength caused a reduction in Δratio. The alginate sharply impacted the Δratio of V_H_H_HER2_. A clear increase of the Δratio was observed for all the assessed pH at an ionic strength of 50 mM. Moreover, increasing the solution ionic strength led to a strong drop of the V_H_H_HER2_ Δratio. In contrast, the effect of the solution pH was fairly limited in the presence of alginate. These observations pointed out the fundamental influence of the alginate on the V_H_H_HER2_ ability to refold. V_H_H_HER2_ solutions that included P407 were clearly influenced by pH and ionic strength. Increasing the ionic strength and decreasing the pH of V_H_H_HER2_ solutions sharply decreased the Δratio. Finally, the pNIPAAM solutions at pH 6 and ionic strengths of 50 or 150 mM were the conditions with the lowest values.

### 3.2. Sustained Release of V_H_H_HER2_ from a Next-Generation Implant

#### 3.2.1. Characterization of 3D-Printed Implant

In this study, a 3D-printed implant was designed and fabricated, as illustrated in Figure 4. The PLA implant was printed as shown in Figure 4A. The implant was a hollow cylinder with one closed end. The inner and outer diameters were, respectively, 2.3 and 3.3 mm, and the length was 32 mm (Figure 4B).

On SEM images, the open end had a roughly regular appearance (Figure 4C). Indeed, regular striations on the interior and exterior could clearly be distinguished which corresponded to the selected layer height of 0.1 mm (Figure 4D–F). The exterior of the implant had a regular, homogeneous structure. The DSC experiments that were performed on the PLA implants served to evaluate the state of the polymer extrudate (Figure 4G). This assessment was carried out by determining the Tg and Tm of three different PLA implants. The thermogram showed two main endothermic events. The first one was the Tg, which began and reached its midpoint at, respectively, 47.3 ± 1.4 and 50.5 ± 1.2 °C. The second one was the Tm of the PLA at 147.5 ± 0.8 °C. The Tg was similar for the three implants, suggesting that the printing process was reproducible.

#### 3.2.2. Hydrogel Homogeneity

The homogeneity of sdAb concentration within the polymer solutions was assessed by fluorescence monitoring. To that end, the mean fluorescence intensity at 330 nm (F330i) was measured at between 20 and 22 °C in the three sdAb solutions which contained pNIPAAM, alginate, and P407. The F330i of BIC25 was slightly increased in the presence of pNIPAAM and alginate in contrast to P407 solutions (Figure 5A). The variation in F330i (as indicated by standard deviation) was higher with the alginate and P407 solutions than with the pNIPAAM solutions. Similarly, the pNIPAAM and alginate solutions were associated with a sharp increase in the F330i of V_H_H_EGFR1_ in comparison to P407 solutions (Figure 5B). The variation in F330i (as indicated by standard deviation) was higher with the P407 and alginate solutions than with the pNIPAAM solutions. Finally, the F330i of V_H_H_HER2_ was almost identical for the three polymers. Surprisingly, the variation in the F330i of V_H_H_HER2_ was relatively small except in the alginate solutions at pH 6 and 50 or 300 mM ionic strengths (Figure 5C).

#### 3.2.3. Release Profile of V_H_H_HER2_ from Next-Generation Implant

The release of V_H_H_HER2_ was evaluated in the alginate and P407 solutions at pH 6, 7, and 8 at a fixed ionic strength of 50 mM. The cumulative percentage of V_H_H_HER2_ release (Figure 6A) was measured in the alginate hydrogels as a function of time. All alginate hydrogels displayed the same two-phase release profile: an initial burst release between 0 and 24 h followed by a slow release between 24 and 312 h.

The alginate hydrogels at pH 6, 7, and 8, respectively, released 35 ± 6%, 32 ± 12%, and 31 ± 2% of V_H_H_HER2_ over the first 24 h. Large variations in the cumulative release percentage were observed at each sampling time point. Obviously, these variations were due to the substantial variation in V_H_H_HER2_ concentration at 24 h (Figure 6B). Unlike the first sampling time points, the concentration of each condition at 144 and 312 h was similar, and thus yielded smaller standard deviations. The V_H_H_HER2_ concentration at 312 h for the alginate solutions at pH 6, 7, and 8 was, respectively, 9 ± 1, 8 ± 0, and 9 ± 1 µg/mL. Finally, V_H_H_HER2_ release from the alginate hydrogels at pH 6, 7, and 8 was terminated after 312 h, the final release percentages being 84 ± 8%, 80 ± 13%, and 81 ± 1%. The V_H_H_HER2_ release profile from the P407 hydrogels also comprised an initial burst release between 0 and 24 h followed by a slow release between 24 and 312 h (Figure 6C). The P407 hydrogels at pH 6, 7, and 8 released 32 ± 1%, 30 ± 0%, and 37 ± 1% of the V_H_H_HER2_ over the first 24 h. This burst release was clearly reflected by the increase in nanobody concentration at 24 h in the P407 hydrogels at pH 6, 7, and 8, which was 16 ± 1, 15 ± 0, and 18 ± 0 µg/mL (Figure 6D). Finally, V_H_H_HER2_ release from the P407 hydrogels at pH 6, 7, and 8 was terminated after 312 h, the final release percentages being 80 ± 4%, 80 ± 1%, and 89 ± 1% (Figure 6D).

To assess possible alterations of the tertiary structure, Δratio analysis was performed on the V_H_H_HER2_ released. Figure 6E demonstrates the fundamental influence of pH on the reversibility of V_H_H_HER2_ unfolding in the alginate hydrogels. The only condition with a Δratio similar to the control condition was the alginate hydrogel at pH 8. In addition, a slight increase in Δratio was noted over time as V_H_H_HER2_ was released from the alginate hydrogels at pH 6 and 7. The F350/330 of V_H_H_HER2_ was slightly altered after extended exposure to the alginate hydrogels (Appendix A), the transition becoming less marked and more linear. Nevertheless, the curve’s appearance was practically unaffected by the increased duration of contact between the nanobody and the alginate. Figure 6F shows the remarkable influence of the P407 hydrogels at pH 6, 7, and 8 on V_H_H_HER2_ Δratio. The F350/330 values were strongly impacted by the extended exposure of V_H_H_HER2_ to the P407 hydrogels (Appendix A).

## 4. Discussion

Our study aimed to confirm the feasibility of a 3D-printed implant combined with a hydrogel for sustaining the release of sdAbs. Using this device, V_H_H_HER2_ release was extended up to 13 days (312 h) with an initial burst followed by a continuous, linear release phase. Our chosen strategy for sustaining sdAb release appears to offer a versatile solution given that V_H_H_HER2_ release was comparable from two different hydrogels based on conventional polymers (alginate and P407). These two hydrogels have different gelation mechanisms and different potential release mechanisms. P407 gelation is based on the formation of micelles and their ordered packing [27], while alginate gelation is based on electrostatic interactions between alginate chains and both Ca^2+^ and V_H_H_HER2_ [28]. In addition, V_H_H_HER2_ release from P407 was probably triggered by protein diffusion and dissolution of the P407 matrix [21], whereas its release from the alginate hydrogels may have been due to complex dissociation between V_H_H_HER2_ and alginate [28,29]. Similar to the P407 hydrogels, V_H_H_HER2_ release from the alginate hydrogels was extended up to 312 h, although with slightly more variations during the first 24 h. These variations were probably due to alginate heterogeneity and initial inhomogeneous crosslinking by Ca^2+^ ions. Despite some assumed differences between these two hydrogels, similar release kinetics were achieved, suggesting that the implant design, and thus surface exchange, played a primary role. The implant design controls the surface exchange between the CRS and the release medium. In our study, an implant was designed around the Implanon^®^ implant [23] and was successfully printed with a well-defined surface morphology. PLA layer superimposition caused by the 3D printing technology was clearly apparent on the inner and outer surfaces but disappeared where the implant was sectioned. Surface uniformity and continuity were fundamental for controlling surface exchange and thus extending sdAb release. Moreover, extrusion did not alter the physical state of the implant, as DSC thermograms showed a Tg and a Tm indicating a partial amorphization of the PLA [30].

Apart from CRS modularity, our chosen strategy also showed itself to be a potential solution for improving patient compliance and acceptance. An advantage of our device is that it may reduce hospitalization and repeated injection sessions, decrease adverse effects by improving sdAb concentration control, and make it possible to quickly remove the device from its subcutaneous site if severe adverse effects occur [23]. Recently, the EMA approved Blincyto^®^ from AMGEN, a bispecific fusion protein of two different scFv which is used to treat acute lymphoblastic leukemia. The mean half-life of Blincyto^®^ is 2.11 ± 1.42 h, and it is administered as a continuous intravenous infusion at 12.5 µg/mL over 28 days with the infusion bag being replaced every 24 h. The dosage for patients >45 kg is split into two main phases, namely 9 µg/day for the first seven days followed by 28 µg/day for the last 28 days [31]. The concentration of Blincyto^®^ is similar to that of the V_H_H_HER2_ released from our CRS. The total dose required is nonetheless greater than the amount of V_H_H_HER2_ loaded into the hydrogel, but it appears a realistic objective. However, Cablivi^®^, a humanized bivalent nanobody, highlights a clear limitation of our strategy in relation to total dose required. This nanobody was recently approved by the EMA to treat acquired thrombotic thrombocytopenic purpura and the dosage is a daily subcutaneous injection of 10 mg at 10 mg/mL over 30 days [32]. In this case, the nanobody payload inside the implant would be very large and therefore seems unrealistic. Consequently, sustained release with our CRS is probably limited to highly potent sdAbs. Additionally, these two marketed sdAbs are stored as lyophilized powder for long-term storage, suggesting that there may be stability issues with the liquid form. This illustrates the essential role of formulation. For instance, the in-use stability of reconstituted Blincyto^®^ and Cablivi^®^ is 96 h at temperatures below 27 °C and 4 h at 25 °C, respectively [31]. Therefore, formulation screening could be one way of foreseeing the effect of our device on sdAb stability and of possibly finding better formulations.

The homogeneity of sdAb concentration within the hydrogel matrix is a fundamental parameter in preparation consistency. Recently, antigen concentration in vaccine formulations was assessed through fluorescence intensity due to its relationship with protein concentration [33]. In this way, the monitoring of intrinsic fluorescence was helpful for assessing the homogeneity of sdAb concentration within polymer solutions. The evaluation looked at fluorescence intensity at 330 nm, and more particularly at variations in the measured values between samples since these could correlate with variations in sdAb concentration or tertiary structure. This assessment of the different hydrogels underlined the primary relationship between the nature of the sdAb and polymer. BIC25 and V_H_H_EGFR1_ in alginate and P407 solutions exhibited significant variations in fluorescence intensity at all pH levels and ionic strengths, but not in pNIPAAM. Interestingly, the distribution of the V_H_H_HER2_ fluorescence values around the mean was relatively tightened, suggesting comparable V_H_H_HER2_ concentration in each sample from the different polymer solutions. The reasons behind the inhomogeneous fluorescence may have lain in solution viscosity, protein–polymer interaction, tertiary structure modification, or protein aggregation. Homogeneous fluorescence between samples is of clear benefit for the subsequent release of the sdAb from the hydrogel and more particularly for obtaining a continuous release over time.

Polymers are occasionally used in protein formulations to stabilize the protein as well as to prevent protein adsorption and protein–protein interactions. A polymer’s effect on a protein depends on the polymer’s nature, more particularly on its amphiphilicity and charge. Nonetheless, the usual polymer concentration is relatively low, at 0.1–1% *w*/*v*, while in sustained release devices it is substantially higher [34].

As detailed previously, sdAb stability is one of the main requirements for the success of the CRS. The prolonged duration of contact between the sdAb and the components of the CRS could lead to protein degradation—for example, loss of tertiary structure—and eventually to aggregation [35,36]. Formulation screening using sdAb fluorescence and backscattering could be used to find formulations that promote sdAb stability and enhance the range of use of our CRS [37,38,39,40]. The solution’s composition, including pH, ionic strength, and choice of polymer, strongly altered sdAb thermal stability. In the absence of polymer, the Tmapp of BIC25 and V_H_H_EGFR1_ was mainly regulated by the solution’s pH and notably increased at pH 6 and 8, respectively. The lack of influence of pH on V_H_H_HER2_ thermal stability during heating was counterbalanced by its fundamental role during cooling. The reversibility of V_H_H_HER2_ unfolding appeared to be clearly affected by pH. The backscattering analysis confirmed that the irreversibility of BIC25 and V_H_H_EGFR1_ unfolding related to aggregation, whereas an absence of aggregation could explain the reversibility of V_H_H_HER2_ unfolding. These analyses of sdAb thermal stability represent a starting point for improving our knowledge of the polymer effect on the sdAb thermal stability.

The pNIPAAM showed a limited influence on sdAb thermal stability which mainly depended on pH. Moreover, the lower fluorescence variation was obtained with the pNIPAAM. Nonetheless, the shape of the F350/330 curves altered sharply in the presence of pNIPAAM, as a second transition appeared close to 45 °C for V_H_H_EGFR1_ and 55 °C for V_H_H_HER2_. This additional V_H_H_EGFR1_ transition was already visible in the backscatter signal of the solutions without polymer. The additional V_H_H_HER2_ transition in the presence of pNIPAAM may also correspond to a variant of V_H_H_HER2_ identified in a previous study [25]. What is more, the BIC25 F350/330 curve was not modified by the presence of pNIPAAM. This last observation demonstrates that the destabilizing effect of the pNIPAAM was protein-dependent and probably limited to protein variants. Regarding the major transition, the pNIPAAM showed little effect on sdAb thermal stability. The Tmapp of BIC25 and V_H_H_EGFR1_ in the presence of pNIPAAM highlighted again the fundamental influence of the solution’s pH. The Tmapp of V_H_H_HER2_ was practically independent of the environmental conditions.

The alginate effect probably arose from interactions between the alginate and sdAbs, as demonstrated by the fundamental role of the solution’s pH and ionic strength. This alginate effect is the consequence of two interlinked factors, namely the net surface charge of the protein as governed by pH, and shielding of electrostatic interactions as governed by ionic strength [28]. Hence, the Tmapp of sdAbs in the presence of the alginate decreased marginally and seemed to be potentiated by the solution’s pH and ionic strength. The Tmapp values of V_H_H_EGFR1_ and BIC25 were mostly affected by pH, whereas those of V_H_H_HER2_ were greatly influenced by ionic strength. BIC25, V_H_H_EGFR1_, and V_H_H_HER2_ have an isoelectric point (pI) of 6.23, 6.57, and 9.23, respectively. First, BIC25 was almost solely affected by the pH, probably due to its net negative surface charge. BIC25 was not able to interact with the alginate and thus its tertiary structure was practically unaltered by electrostatic interactions with alginate. Second, the V_H_H_EGFR1_ Tmapp decreased in the solutions with low ionic strength at pH 6 and 7 but not in the solution at pH 8. The V_H_H_EGFR1_ had a net positive charge in solutions at pH 6, and this net positive charge decreased as pH increased. The positive charge of V_H_H_EGFR1_ probably enabled interaction with the negative charge of the alginate chains. This interpretation was supported by the observed decrease in the effect of ionic strength on V_H_H_EGFR1_ as pH increased from 6 to 8. Third, the Tmapp of V_H_H_HER2_ was mainly dependent on ionic strength and practically independent of pH. Indeed, V_H_H_HER2_ had a net positive charge at all the pH levels assessed and could therefore interact with the net negative charge of the alginate polymer. These electrostatic interactions could alter thermal stability by destabilizing tertiary structure [41]. Hence, the shielding of electrostatic interactions caused by a solution’s ionic strength could be why the Tmapp increased as ionic strength increased. Furthermore, the influence of ionic strength on the Tmapp values of V_H_H_EGFR1_ and V_H_H_HER2_ was relatively nonexistent in the solutions without polymers, and thus may have been directly connected with the presence of alginate.

The effect of P407 on thermal stability appears to be linked to the nature of the sdAb and to the solution’s ionic strength. The Tmapp values of the nanobodies highlighted a mostly ionic-strength-dependent negative influence of P407 on thermal stability. Interestingly, the effect of ionic strength was practically nonexistent in nanobody solutions without polymers. A possible explanation is that ionic strength—and more particularly sodium chloride concentration—influences P407 hydrogel formation. Increasing the salt concentration decreases the gelation temperatures of P407 hydrogels [42] and strengthens the hydrogel matrix due to strong crosslinking of sodium salt with the poloxamer [43]. Nevertheless, the Tmapp of BIC25 was practically unaffected by P407, being instead primarily influenced by the solution’s pH. Evaluating the polymer effect through Tmapp analysis provided an insight into sdAb thermal stability and potential interactions between the sdAb and polymer. However, evaluating the long-term stability of sdAbs requires more in-depth analysis which takes account of the particularities of the sdAbs so as to improve methodological predictability.

Predicting the long-term stability of proteins during processing and storage is a huge challenge. Stressed and accelerated stability studies are frequently used initially to identify which formulations are most likely to maximize protein stability in real-time stability studies. Nonetheless, these stressed thermal stability studies do not fully reflect all the stresses encountered by the protein during processing and storage, and using them as predictive tools implies that many assumptions must be met. Tm is a frequently used thermodynamic parameter known to quantify reversible unfolding and kinetics. Nanobodies emerged as single-domain antibodies with unique features, especially supposedly reversible thermal unfolding [17,44]. Nevertheless, Kunz et al. demonstrated that nanobodies generally unfold irreversibly on thermal gradient signifying that reversible unfolding is the exception in nanobodies. Thus, these authors highlighted the utility of Tm shifts in predicting the aggregation propensity of nanobodies. Briefly, nanobody aggregation alters the unfolding equilibrium between folded and unfolded species. This disruption is caused by unfolded species consumption in the aggregation pathway. According to Le Chatelier’s principle, increasing the nanobody concentration enhances the formation of unfolded species and thus increases aggregation, which in turn reduces Tmapp [26,45]. ΔTmapp was used to assess the effect of formulation on the thermal stability of nanobodies, whereas Tmapp analysis is based on the equilibrium between folded and unfolded species and does not take aggregation into account. Consequently, the conclusions of the ΔTmapp approach should be preferred, bearing in mind that Tm shift can with certain nanobodies be unpredictive [45]. Moreover, the ΔTmapp method could improve our comprehension of polymer effect on the thermal stability of V_H_H_EGFR1_. pNIPAAM was the only polymer to be subjected to ΔTmapp analysis due to the inhomogeneity of the P407 and alginate hydrogels. The ΔTmapp of V_H_H_EGFR1_ was considerably affected by pH. ΔTmapp decreases were associated with a shift in pH toward acidic values, suggesting reduced V_H_H_EGFR1_ aggregation. This trend was observed with or without pNIPAAM, indicating limited interaction between pNIPAAM and V_H_H_EGFR1_. In contrast, positive ΔTmapp values in solutions at pH 6 may have originated in an absence of aggregation and an effective stabilization of the native species, potentially due to an increase in the ratio of sdAb to polymer. Nevertheless, preliminary requirements and correlation studies concerning monomer loss should be performed to confirm the applicability and predictability of this approach in the presence of a polymer. Recently, the reversibility of nanobody unfolding was explored using the Δratio to confirm the extent of structure recovery. Δratio analysis demonstrated its ability to distinguish between V_H_H_HER2_ formulation conditions. For instance, the magnitude of F350/330 transition was shown to correlate primarily with pH, and less so with ionic strength. Concerning the polymers, their capacity to influence the reversibility of unfolding depended on the type of polymer, and was modulated by the solution’s pH and ionic strength. Firstly, pNIPAAM did not modify the reversibility of V_H_H_HER2_ unfolding. In fact, the Δratio values were similar in the presence and absence of pNIPAAM, and the trend concerning pH and ionic strength was almost identical. Secondly, alginate demonstrated a capacity to interfere with V_H_H_HER2_ structure recovery. This modulation of structure recovery was strongly influenced by ionic strength, which, when increased, led to a shielding of electrostatic interaction, and thus enabled V_H_H_HER2_ structure recovery. This fundamental effect of ionic strength suggests potential interaction between the alginate and V_H_H_HER2_, as previously described [28]. P407 had a similar effect on the reversibility of V_H_H_HER2_ unfolding as pNIPAAM. The Δratio values were close to those of V_H_H_HER2_ solutions without polymer, indicating limited interaction with P407.

## 5. Conclusions

Combining a 3D-printed implant with a hydrogel successfully sustained the release of sdAbs, resulting in an initial burst release and a continuous release up to 312 h. Using two different hydrogels, the versatility of the CRS was also demonstrated. Moreover, biochemical analysis improved our knowledge of our CRS, increasing the probability of successfully releasing stable, active sdAb. Interaction between sdAb and CRS potentially determined the success of such approach as interactions within drug delivery device. Therefore, exploring sdAb stability through screening approach is helpful in determining the most suitable formulation conditions. Screening in the form of accelerated stability studies, especially using thermal gradients, requires a thorough understanding of the basis and limits of such an approach. In this study, we succeeded in applying a method that took account of the particularities of sdAbs, such as the reversibility (Tmapp and Δratio) and irreversibility (ΔTmapp) of unfolding. More in-depth characterization was achieved, improving the predictability of screening. This enables those working on the formulation to gain greater certainty, potentially save time and resources, and help mitigate the risks inherent to development by improving the probability of success.

## Figures and Tables

**Figure 1 pharmaceutics-12-00921-f001:**
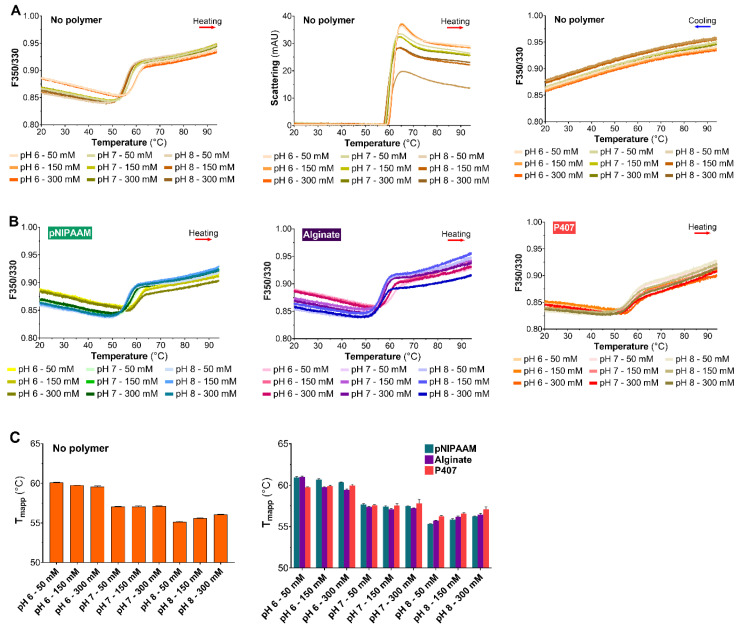
Thermal stability of BIC25 (0.5 mg/mL): (**A**) wavelength fluorescence intensity ratio between 350 and 330 nm (F350/330) (left) and light backscatter signal (middle) during the heating ramp and F350/330 during the cooling ramp (right) at different pH and ionic strength conditions (color legend in the figure); (**B**) F350/330 in the presence of 8 mg/mL pNIPAAM (left), 8 mg/mL alginate (middle) and 200 mg/mL P407 (right) at different pH and ionic strength conditions (color legend in the figure); and (**C**) apparent midpoint of thermal denaturation at different pH and ionic strength conditions (left) and apparent midpoint of the thermal denaturation of BIC25 in the presence of poly(N-isopropylacrylamide) pNIPAAM (green), alginate (purple) and Poloxamer 407 (red) at different pH and ionic strength conditions (middle). The results are presented as mean ± standard deviation (*n* = 4).

**Figure 2 pharmaceutics-12-00921-f002:**
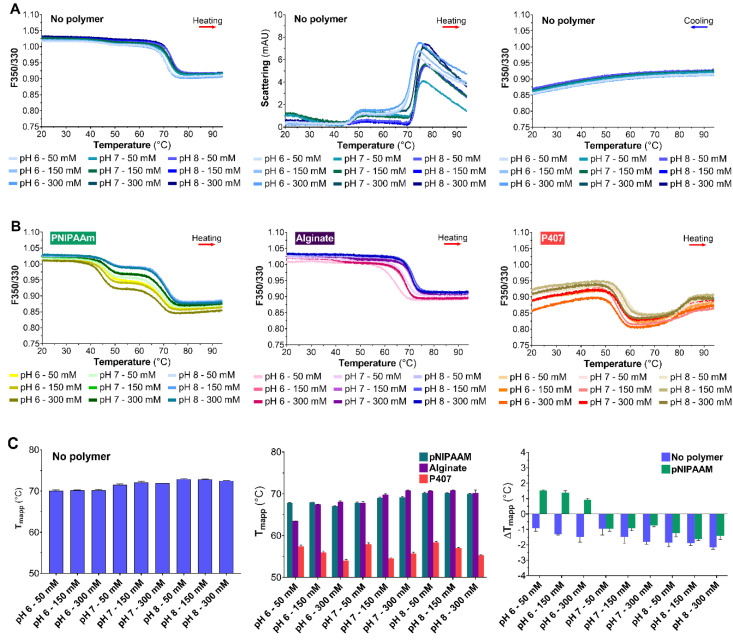
Thermal stability of V_H_H_EGFR1_ (0.5 mg/mL): (**A**) F350/330 (left) and light backscatter signal (middle) during the heating ramp and F350/330 during the cooling ramp (right) at different pH and ionic strength conditions (color legend in the figure); (**B**) F350/330 in the presence of 8 mg/mL pNIPAAM (left), 8 mg/mL alginate (middle) and 200 mg/mL P407 (right) at different pH and ionic strength conditions (color legend in the figure); and (**C**) apparent midpoint of thermal denaturation at different pH and ionic strength conditions (left), apparent midpoint of the thermal denaturation of V_H_H_EGFR1_ in the presence of pNIPAAM (green), alginate (purple), and P407 (red) at different pH and ionic strength conditions (middle), and ΔTmapp of V_H_H_EGFR1_ between 0.5 and 0.2 mg/mL with or without pNIPAAM (right). The results are presented as mean ± standard deviation (*n* = 4).

**Figure 3 pharmaceutics-12-00921-f003:**
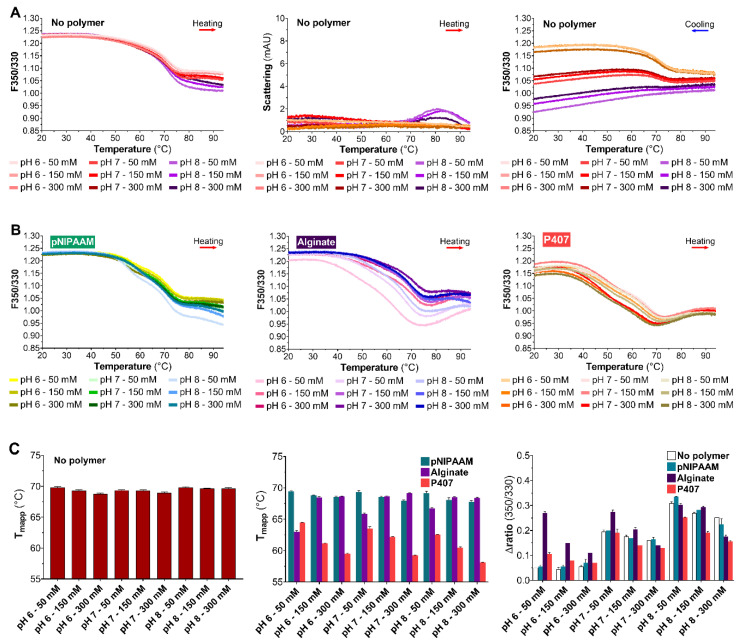
Thermal stability of V_H_H_HER2_ (0.5 mg/mL): (**A**) F350/330 (left) and light backscatter signal (middle) during the heating ramp and F350/330 during the cooling ramp (right) at different pH and ionic strength conditions (color legend in the figure); (**B**) F350/330 in the presence of 8 mg/mL pNIPAAM (left), 8 mg/mL alginate (middle), and 200 mg/mL P407 (right) at different pH and ionic strength conditions (color legend in the figure); and (**C**) apparent midpoint of thermal denaturation at different pH and ionic strength conditions (left), apparent midpoint of the thermal denaturation of V_H_H_HER2_ in the presence of pNIPAAM (green), alginate (purple), and P407 (red) at different pH and ionic strength conditions (middle), and difference between the initial and final ratios of the heating and cooling ramps (Δratio) without polymer (white), with pNIPAAM (green), with alginate (purple), and with P407 (red) at different pH and ionic strength conditions (right). The results are presented as mean ± standard deviation (*n* = 2).

**Figure 4 pharmaceutics-12-00921-f004:**
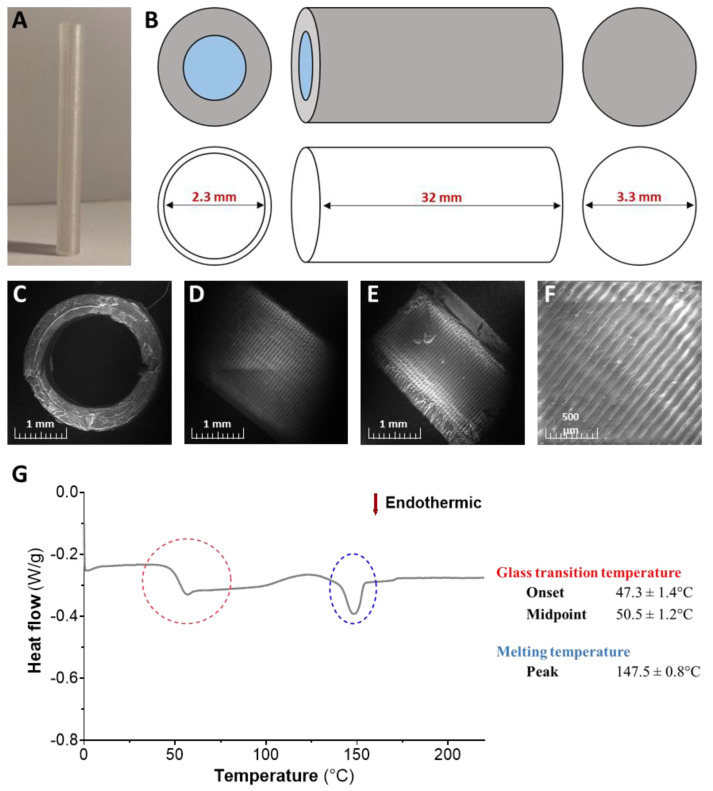
Characteristics of the 3D-printed PLA implant. Photograph of the 3D-printed PLA implant (**A**) and diagram of its dimensions (**B**). Scanning electron microscopy images of the implant’s open end (**C**), outer surface (**D**), and inner surface (**E**,**F**). Differential scanning calorimetry profiling of the 3D-printed implant (**G**) was carried out for illustrative purposes and determined the polymer’s melting and glass transition temperatures. The results are presented as mean ± standard deviation (*n* = 3).

**Figure 5 pharmaceutics-12-00921-f005:**
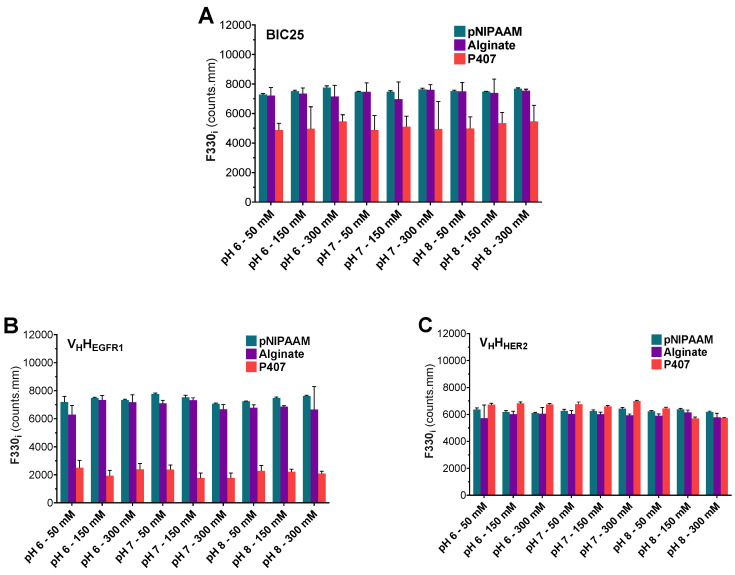
Mean fluorescence intensity of sdAbs. Mean fluorescence intensity at 330 nm between 20 °C and 22 (F330i) of 0.5 mg/mL BIC25 (**A**), V_H_H_EGFR1_ (**B**), and V_H_H_HER2_ (**C**) in the presence of 8 mg/mL pNIPAAM (green), 8 mg/mL alginate (purple), and 200 mg/mL P407 (red) in solutions with defined pH and ionic strength. The results are presented as mean ± standard deviation (*n* = 4).

**Figure 6 pharmaceutics-12-00921-f006:**
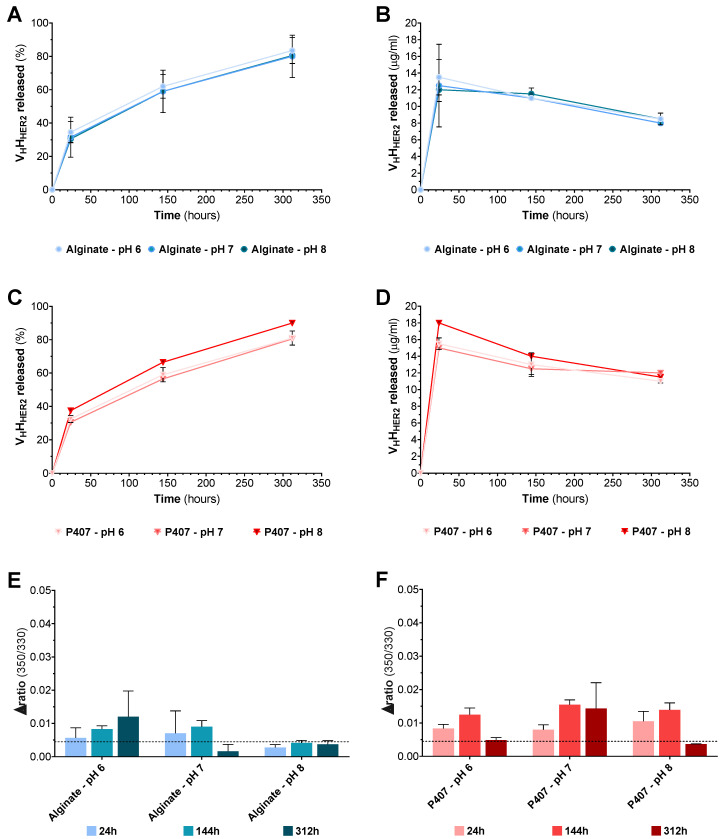
Release and stability profiles of V_H_H_HER2_. Cumulative release percentages (**A**) and concentration (**B**) of V_H_H_HER2_ released from alginate hydrogels from loaded implants in the release medium at each sampling time point. Cumulative release percentages (**C**) and concentration (**D**) of V_H_H_HER2_ released from P407 hydrogels from loaded implants in the release medium at each sampling time point. Δratio of the heating and cooling ramps of V_H_H_HER2_ from alginate (**E**) and P407 (**F**) hydrogels. The dotted line represents the Δratio of the control V_H_H_HER2_ (12.5 µg/mL) in the release medium.

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
