# Peer review of "Contribution of Intrinsic Fluorescence to the Design of a New 3D-Printed Implant for Releasing SDABS"

_pharmaceutics, 2020, doi:10.3390/pharmaceutics12100921_

Round 1

Reviewer 1 Report

The paper is interesting, well organized and the work described therein well performed. The paper would benefit if in the discussion section the proposed fluorescence-based methods proposed for assessing the stability of the sdAbs in the polymer solutions were critically assessed and compared to other available methods  for assessing protein stability.

Specific Comments:

Line 21: “Through the screening step, we gained knowledge about the influence of the choice of polymer and about potential interactions between the sdAb and the polymer.” How exactly this knowledge was used in the design/formulation of the proposed delivery system?

Line 17: “The results demonstrated the implant’s ability to sustain sdAb release up to 13 days through a reduced initial burst release followed by a continuous release.” Reduced burst release compared to what? Is a burst release of 30% (1/3 of sdAb content of the delivery system) in 24 hours satisfactory?

Can the release be regulated, i.e. adapted to specific medical sdAb delivery needs, and how? How is sdAb loading (content) in the hydrogel affects release rate?

The experimental procedure for measuring hydrogel homogeneity is not given in the experimental section.  Apparently, the homogeneity was assessed  by SD of fluorescence measurements (Fig. 5). The “homogeneity” in the absence of the polymer should be provided as control.  

Line 558: “The pNIPAAM showed a limited influence on sdAb thermal stability which mainly depended on pH. Surprisingly, the pNIPAAM also revealed a potential lack of nanobody homogeneity within the hydrogel.” But the lower fluorescence variation was obtained with the pNIPAAM (Fig. 5), thus the statement is not substantiated by available data.  

Line 649: "Combining a 3D-printed implant with a hydrogel successfully sustained the release of sdAbs, 649 resulting in almost no burst release..." The conclusion is not substantiated by the obtained release profiles (Fig. 6).

Author Response

Point 1 (Line 21): “Through the screening step, we gained knowledge about the influence of the choice of polymer and about potential interactions between the sdAb and the polymer.”

How exactly this knowledge was used in the design/formulation of the proposed delivery system?

Response 1

The screening study highlighted that each combination between antibody fragment and polymer should be assessed as a special case. Through the screening step, we observed that a polymer could have a negative effect on thermal stability but enables a satisfactory protein dispersion within hydrogel matrix (VHHEGFR1/pNIPAAM), and vice versa. Based on the screening experiments, VHHHER2 has been chosen for the release experiments due its good homogeneity in all the assessed polymer solutions. Intrinsic fluorescence appears as an efficient method for the evaluation of a sdAb-polymer combination.

Point 2 (Line 17): “The results demonstrated the implant’s ability to sustain sdAb release up to 13 days through a reduced initial burst release followed by a continuous release.”

Reduced burst release compared to what? Is a burst release of 30% (1/3 of sdAb content of the delivery system) in 24 hours satisfactory?

Response 2

We agree with the reviewer’s observation that a 30% burst release in 24h could be considered as not satisfactory. Nonetheless, a burst release of 30% is relatively classic for a hydrogel for sustaining protein release (Bae et al – 2015; Bao et al – 2020; Delgado et al – 2002; Delgado et al – 2015 ; Fan et al – 2020).

Our work is quite difficult to compare with literature data. Welty number of publications reported the use of whole antibodies in controlled release systems. And in other part, the nature of polymer and nature of crosslinking are often different. According to these considerations, we can consider that the obtained burst-release is comparable to the other systems to some extent. Consequently, we have revised all the sentences with the term “reduced/no burst release” and changed the sentence as follows: “an initial burst release” throughout the revised version of the manuscript.

References:

  • Bae et al – 2015 (https://pubmed.ncbi.nlm.nih.gov/26100344/)
  • Bao et al - 2020 (https://www.sciencedirect.com/science/article/abs/pii/S0141813020327963)
  • Delgado et al – 2002 (https://pubs.acs.org/doi/abs/10.1021/bm010147y)
  • Delgado et al – 2015 (https://www.nature.com/articles/srep12374)
  • Fan et al – 2020 (https://www.sciencedirect.com/science/article/abs/pii/S0927776520306172)

Point 3: Can the release be regulated, i.e. adapted to specific medical sdAb delivery needs, and how? How is sdAb loading (content) in the hydrogel affects release rate?

Response 3

Many thanks to you for this suggestion. Common sense consideration allows us to suggest that the release could be modulated by the implant composition or structure. As an example, a 3D printed implant made of PLA and calcium chloride could lead to a more homogeneous crosslinking of alginate hydrogel and thus could modulate the antibody fragment release. The implant shape, and more particularly the size and dimension of the open part, could modify the antibody fragment release by increasing or decreasing exchange with the release medium.

In the first part of discussion section, we discussed the potential of our CRS regarding Blincyto® dose and concentration. We observe that our device had a release kinetic that potentially matches with the therapeutic need of Blincyto®.

Finally, the law of diffusion (Fick’s law) allows us to think that increasing the protein loading will probably result in a faster release kinetic.

Point 4: The experimental procedure for measuring hydrogel homogeneity is not given in the experimental section. Apparently, the homogeneity was assessed by SD of fluorescence measurements (Fig. 5). The “homogeneity” in the absence of the polymer should be provided as control.

Response 4

As suggested by the reviewer, we have added a section in the Materials & Methods part describing the assessment of hydrogel “homogeneity”.

Moreover, we also proposed a figure that includes the “homogeneity” in the absence of polymer. Should we replace the figure 5 by this present figure in the revised form of the article?

(Figures avalaible on pdf document)

Point 5 (Line 558): “The pNIPAAM showed a limited influence on sdAb thermal stability which mainly depended on pH. Surprisingly, the pNIPAAM also revealed a potential lack of nanobody homogeneity within the hydrogel.” But the lower fluorescence variation was obtained with the pNIPAAM (Fig. 5), thus the statement is not substantiated by available data.

Response 5

Thank you for pointing this out. The reviewer is correct, and we propose to reformulate the sentence as follows: “The pNIPAAM showed a limited influence on sdAb thermal stability which mainly depended on pH. Moreover, the lower fluorescence variation was obtained with the pNIPAAM”. This sentence has been included in the revised form of the article.

Point 6 (Line 649): "Combining a 3D-printed implant with a hydrogel successfully sustained the release of sdAbs, resulting in almost no burst release..." The conclusion is not substantiated by the obtained release profiles (Fig. 6).

Response 6

We agree with the reviewer’s assessment. Accordingly, we have revised the sentence throughout the manuscript and proposed to change the sentence as follow: ”Combining a 3D-printed implant with a hydrogel successfully sustained the release of sdAbs, resulting in an initial burst release...". This sentence has been included in the revised form of the article.

Reviewer 2 Report

This manuscript presents a good research on 3D-printed protein formulation. The authors designed a combined implant with a 3D-printed PLA device and the SDABS-loaded hydrogels filling in this device. This implant could sustained release SDABS for 13 days. This work is a useful exploration of the application of 3D printing in drug delivery systems. It is clearly written. I recommended it to be published on Pharmaceutics after revision. The following points shoud be addressed in the revised manuscript. I am not an expert in protein formulation, so leave in-depth analysis of this to others.

1. Page 16, Line 498-499. The authors concluded that …as DSC thermograms showed a Tg and a Tm indicating no amorphization of the PLA. First, as well known, PLA is a semi-crystalline polymer. Second, Figure 4 shows that there is a big and broad exothermic peak before the melting of PLA occurs. Therefore, it is inappropriate to conclude there is no amorphization of the PLA during extrusion based on the observation of glass transition and melting events.

2. The external diameter of the printed device is 3.3 mm, which is not small for an implant. Please comment on this.

3. Some typographical or related errors.

1) Page 1, Line35. Please delete the number “1” at the end of the word inflammation.

2) Page 5, Line 190. Please correct “2.5.13. D Printing” to “2.5.1. 3D printing”.

3) In Figure 1-3, the front size of the words is too small.

Author Response

Point 1 (Line 498-499): The authors concluded that …as DSC thermograms showed a Tg and a Tm indicating no amorphization of the PLA. First, as well known, PLA is a semi-crystalline polymer. Second, Figure 4 shows that there is a big and broad exothermic peak before the melting of PLA occurs. Therefore, it is inappropriate to conclude there is no amorphization of the PLA during extrusion based on the observation of glass transition and melting events.

Response 1

We agree that the term “amorphization” was inappropriate and our interpretation goes a little bit further. We propose to change the sentence as follows: ”…as DSC thermograms showed a Tg and a Tm indicating a partial amorphization of the PLA” in the revised version of the manuscript.

Point 2: The external diameter of the printed device is 3.3 mm, which is not small for an implant. Please comment on this.

Response 2

Thank you for pointing this out. As noticed by the reviewer, the external diameter is in the range of the commercially developed implants such as DUROS® and IMPLANON® (Kleiner et al – 2014; Wright et al – 2001; Vaishya et al - 2015). One of the main limitations of this device is to achieve a sufficient protein loading to allow an adequate release duration. We would like to increase the implant size for increasing hydrogel volume and thus protein loading. Nevertheless, the implant dimensions could be a limit for its acceptance by patients.

Point 3 (Line35): Please delete the number “1” at the end of the word inflammation.

Response 3

The number “1” has been corrected on line 35 in the revised version of the manuscript.

Point 4 (Line 190): Please correct “2.5.13. D Printing” to “2.5.1. 3D printing”.

Response 4

The title “2.5.13. D Printing” has been corrected on line 192 in the revised version of the manuscript.

Point 5 (Figure 1-3): the front size of the words is too small.

Response 5

The font size of the figure 1 – 3 has been change in the revised version of the manuscript.

Reviewer 3 Report

Manuscript entitled "Contribution of Intrinsic Fluorescence to the Design
of a New 3D-Printed Implant for Releasing SDABS" by authors Alexandre Nicolas et al., have made an attempt to design an implant with 3D printing technique for sustained release of single domain antibodies for therapeutic purpose. 

Authors have designed a biocompatible cylinder shape implant loaded with hydrogels and SDABS. Authors have studied the accelerated thermal stability study for the formulations. Finally release pattern of the implants were determined using BCA protein estimation in a vial containing buffers at 37ºC. 

Authors have presented the results in 6 figures. Experimental methodology is provided in detail. 

please address the following comments

  1. Line # 35 - cancer and inflammation [1] Typo error
  2. Line # 105: Typo error- Poly lactic acid (PLA)- please look in to parentheses
  3. Line # 111: please abbreviate BBS
  4. Line # 144 and 145: please re-write the sentence. looks same sentence repeated again.
  5. Please provide reference(s) for thermal stability studies
  6. Line# 190: please correct 3D printing (typo error)
  7. Line # 228: please include "Release Study" also in the subtitle along with BCA assay
  8. Reference 31 and 32 links are not working. please provide valid links.
  9. These findings needs to be validated using in vivo models. what are the suggestions you are planning to propose for in vivo validation of this Design. 
  10. Authors have listed some of the limitations of their formulation. However, Authors can provide the classification/category of antibodies are more suitable for this kind of formulation.
  11. Authors may explain how do the sustained release formulation can increase the biological half life of the SDABS?

Author Response

Point 1 (Line 35): cancer and inflammation [1] Typo error.

Response 1

Thank you for pointing this out. The number “1” has been corrected on line 35 in the revised version of the manuscript.

Point 2 (Line 105): Typo error- Poly lactic acid (PLA)- please look in to parentheses

Response 2

The name “Poly lactic acid” has been corrected on line 105 in the revised form of the manuscript.

Point 3 (Line 111): please abbreviate BBS.

Response 3

The term BBS was replaced by “borate buffered saline” (line 111) in the revised form of the manuscript.

Point 4 (Line 144): please re-write the sentence. looks same sentence repeated again.

Response 4

As suggested by the reviewer, we rewrote the sentence as follows: “The alginate, P407 and pNIPAAM solutions were prepared by dissolving each polymer in a citrate-phosphate solution with last two in ice-bath.” in the revised form of the manuscript.

Point 5: Please provide reference(s) for thermal stability studies.

Response 5

We agree with the reviewer’s comment. We added three references about “thermal stability studies” on line 556 in the revised form of the manuscript.

References:

-          Goldberg et al – 2010 (https://pubmed.ncbi.nlm.nih.gov/20960568/),

-          Magnusson et al – 2018 (https://pubmed.ncbi.nlm.nih.gov/30414312/),

-          Kotov et al – 2019 (https://www.nature.com/articles/s41598-019-46686-8)

-          Boland et al – 2018 (https://pubs.acs.org/doi/10.1021/acs.analchem.8b03176).

Point 6 (Line 190): please correct 3D printing (typo error).

Response 6

The typo error was corrected (line 192) in the revised form of the manuscript.

Point 7 (Line 228): please include "Release Study" also in the subtitle along with BCA assay.

Response 7

The title was changed and now includes the term “Release Study” (line 236) in the revised form of the manuscript.

Point 8 (Ref 31 & 32) : Reference 31 and 32 links are not working. please provide valid links.

Response 8

Many thanks to you for pointing this out. The problem seems to be related to the presence of a point at the end of the reference. We delete this point to allow a direct use of these links.

  • https://www.ema.europa.eu/en/documents/product-information/blincyto-epar-product-information_en.pdf
  • https://www.ema.europa.eu/en/documents/product-information/cablivi-epar-product-information_en.pdf

Point 9: These findings need to be validated using in vivo models. what are the suggestions you are planning to propose for in vivo validation of this Design ?

Response 9

As suggested by the reviewer, an in vivo study could be performed to validate the usefulness of our device. Many models have been used to evaluate the type of implants such as mouse and rat, or even non-human primate models. In our case, the model that seems the most appropriate could be the minipig model. In fact, the subcutaneous tissue of mini-pigs appears to be particularly similar to human tissues (Collins et al – 2017; Kang et al – 2013; Milewski et al – 2015, Zhen et al., 2012)).

References:

  • Collins et al – 2017 (https://pubmed.ncbi.nlm.nih.gov/28707164/)
  • Kang et al – 2013 (https://pubmed.ncbi.nlm.nih.gov/23376811/)
  • Milewski et al – 2015 (https://pubmed.ncbi.nlm.nih.gov/25460581/)
  • Zhen et al., 2012- (https://www.ncbi.nlm.nih.gov/pmc/articles/PMC3361660/)

Point 10: Authors have listed some of the limitations of their formulation. However, Authors can provide the classification/category of antibodies are more suitable for this kind of formulation.

Response 10

As suggested by the reviewer, we could propose a classification of antibody fragments that are more suitable for the application of sustained release device. After this work, we would suggest that a bivalent nanobody could be a better choice instead of monovalent nanobody. Indeed, the bivalent nature allows avidity for its target and thus enhancing tissue penetration. Moreover, the size of a bivalent nanobody could also extend its release from the device due to its slower diffusion in comparison to a monovalent nanobody (Thurber et al - 2008  ).

Reference:

  • Thurber et al – 2008 (https://www.sciencedirect.com/science/article/abs/pii/S0169409X08001233)

Point 11: Authors may explain how do the sustained release formulation can increase the biological half-life of the SDABS?

Response 11

To our opinion, the implant device does not increase sdAb half-life per se, but improves the residence time within the body and in turn, allows a sustained plasmatic concentration that counteracts their limited half-life. In fact, the entrapped antibody fragments in our device are protected from physiological medium which degrades them. After its release, the sdAb has the same half-life as a natural antibody fragment in the body.

Reviewer 4 Report

The authors have designed a novel 3D printed scaffold capable of releasing single domain antibodies and have characterized the release profiles, thermal stability, hydrogel strength of the construct. The work is novel and the characterization has been done in sufficient detail. The dose minimization against compared to administering heavier IgGs is indeed a significant application of sdAbs and hence it is a promising investigation. However the authors can provide more clarity on the below points: 1. The purity/grade, viscosity, guluronate: manuronte ratio of the sodium alginate used in this study should be mentioned in the methods. 2. The promotors/constructs used in the production of sdAbs and fragments in the EColi cells should be mentioned 3. Line 190 : Correct the heading (3D printing) 4. The influence of hydrogel matrices on the ability of the sdAb to fold/ refold in different buffer (pH) conditions and temperature for BIC25, VH_EGFR1 and VH_HER2 has been characterized well by fluorescence spectroscopy 5. It is not clear how the sdABs+hydrogel mixture were loaded onto the 3D printed PLA scaffold. Was it via injection? Hence the uniformity of the gel+drug loading on the scaffold itself is hard to assess. Can the authors include a schematic/explain in more detail. 6. Have authores noticed/measured any changes in the mechanics/stability of the 3D construct after hydrogel loading? 7. The high viscosity alginate could be more stable than pNiPAAm in room temperature (before in situ gelling/) does the hydrogels while inside the 3D printed construct solidify at higher temperature? and is that temperature conducted well by PLA?

Author Response

Response to Reviewer 4 Comments

Point 1: The purity/grade, viscosity, guluronate: manuronate ratio of the sodium alginate used in this study should be mentioned in the methods.

Response 1

As noticed by the reviewer, we have added information about the viscosity of sodium alginate as follows: ”High-viscosity alginate sodium salt derived from brown algae (viscosity: 1320 mPa.s [1% w/v])” (line 103-104). Nonetheless, we do not have any information about the guluronate:mannuronate ratio.

Point 2: The promotors/constructs used in the production of sdAbs and fragments in the EColi cells should be mentioned.

Response 2

Many thanks for your suggestion. The vector for the nanobodies and BIC25 production is a pET-15b and a pCDNA3.4, respectively. The promotor for the nanobodies and BIC25 production was promotor T7 and CMV, respectively. We have added information about the promotors/constructs (line 103-104, 109-110 and 120) in the Materials & Methods section.

Point 3 (Line 190): Correct the heading (3D printing).

Response 3

The heading has been corrected.

Point 4: The influence of hydrogel matrices on the ability of the sdAb to fold/ refold in different buffer (pH) conditions and temperature for BIC25, VH_EGFR1 and VH_HER2 has been characterized well by fluorescence spectroscopy.

Response 4

Thank you.

Point 5: It is not clear how the sdABs+hydrogel mixture was loaded onto the 3D printed PLA scaffold. Was it via injection? Hence the uniformity of the gel+drug loading on the scaffold itself is hard to assess. Can the authors include a schematic/explain in more detail?

Response 5

Thank you for pointing this out. The antibody fragment/polymer mixture was loaded onto the 3D printed PLA with a 1 mL syringe and a needle (0.8 x 50 mm needle). The process starts with a careful insertion of the needle into the implant. Once the needle is at the implant bottom, the filling is initiated with a particular attention that the end of the needle remains at the hydrogel surface. The essential parameters monitored during the filling procedure are the absence of bubbles and the plenty filling of the implant. All these parameters were controlled visually.

 Figure available on pdf document

Point 6: Have authors noticed/measured any changes in the mechanics/stability of the 3D construct after hydrogel loading?

Response 6

Macroscopically, we did not observe any change of the PLA implant after hydrogel loading but mechanical studies could improve the knowledge about hydrogel-implant compatibility.

Point 7 : The high viscosity alginate could be more stable than pNiPAAm in room temperature (before in situ gelling/) does the hydrogels while inside the 3D printed construct solidify at higher temperature? and is that temperature conducted well by PLA?

Response 7

Thank you for pointing this out. We agree with your observation that PLA implant could limit the temperature homogenization. We did not use pNIPAAM for the release study but only P407 and alginate. For the P407, the viscosity of the mixture was already relatively high at room temperature, so we had not doubt about hydrogel formation upon injection in the PLA implant.
